Effects of unstable training on muscle activation: a systematic review and meta-analysis of electromyographic studies

Bao Zihan
Wang Shun 67839417@qq.com
Li Ziyang
School of Physical Education, Huaibei Normal University , Huaibei , China
Robinson Mark
Electronic publication date: 2025 Jul 24
Publication date: 2025
Volume: 13
Electronic Location ID: e19751
Received 2025 Jan 8; Accepted 2025 Jun 24
Copyright: © 2025 Bao et al.
Copyright year: 2025
Copyright holder: Bao et al.
License: This is an open access article distributed under the terms of the Creative Commons Attribution License, which permits unrestricted use, distribution, reproduction and adaptation in any medium and for any purpose provided that it is properly attributed. For attribution, the original author(s), title, publication source (PeerJ) and either DOI or URL of the article must be cited.
License URL: https://creativecommons.org/licenses/by/4.0/

Keywords: Unstable training, Muscle electromyography, Muscle activation, Activation prescription, Meta-analysis

Funding: National Social Science Fund of China 23BTY098 This study was supported by the General Project of the National Social Science Fund of China (No. 23BTY098) The funders had no role in study design, data collection and analysis, decision to publish, or preparation of the manuscript.

==============================
Objective

To systematically evaluate the effects of unstable training (UT) on muscle activation and provide activation prescriptions for different muscle regions, offering more targeted guidance for different populations in muscle activation.

Methods

Data extraction and meta-analysis were conducted using RevManager 5.3, Stata 16.0, and R software. Subgroup analyses were performed on five variables: exercise equipment, exercise intensity, exercise mode, exercise experience, and contraction mode. Heterogeneity and publication bias were also examined.

Results

A total of 28 studies were included, involving 579 participants. Comparison of activation effects between unstable training and stable training: Significant increases in core muscle activation, including rectus abdominis (SMD = 0.32, 95% CI [0.18–0.46], P < 0.01), internal oblique (SMD = 0.38, 95% CI [0.20–0.56], P < 0.01), external oblique (SMD = 0.38, 95% CI [0.20–0.56], P < 0.01), and erector spinae (SMD = 0.60, 95% CI [0.17–1.02], P < 0.01); Significant increases in upper limb muscle activation, including biceps brachii (SMD = 0.52, 95% CI [0.23–0.80], P < 0.01), trapezius (SMD = 0.23, 95% CI [0.12–0.35], P < 0.01), serratus anterior (SMD = 0.33, 95% CI [0.07–0.59], P = 0.01), and triceps brachii (SMD = 0.24, 95% CI [0.04–0.45], P = 0.02); Significant increases in lower limb muscle activation, including soleus (SMD = 0.65, 95% CI [0.42–0.87], P < 0.01), gluteus medius (SMD = 0.28, 95% CI [0.05–0.52], P = 0.02). In subgroup analysis, the core muscles with the great effect were: rectus abdominis (Bosu ball, body weight, sit-ups), internal oblique (Swiss ball, relative load, bench press), external oblique (Swiss ball, body weight, sit-ups), erector spinae (TRX suspension, body weight, bridging); the upper limb muscles with the great effect were: biceps brachii (more than 1 year of training experience, TRX suspension, body weight, muscle-up), trapezius (less training experience, Bosu ball, body weight, push-ups), triceps brachii (body weight). The lower limb muscles with the great effect were: soleus (squats). Negative activation effects: erector spinae (Swiss ball, 60% 1RM load, and shoulder press), serratus anterior (Swiss ball), triceps brachii (more than 1 year of training experience), Swiss ball, >60% 1RM; rectus femoris (Bosu ball, squats).

Conclusion

Unstable training is an excellent choice for rehabilitation after sports injuries, pre-exercise activation, and enhancing specific sports abilities. It can significantly activate core, upper limb, and lower limb muscles. In the future, more rigorous research should be carried out, providing a larger sample size and more meticulous evaluation methods for further comparative analysis.

Introduction

Unstable training (UT) refers to conducting a variety of strength and stability trainings by independently maintaining balance and controlling movements on the created unstable surfaces with the aid of various unstable devices. This training increases the difficulty level, enhances the body’s response to unstable conditions, and thus boosts the activation level of the core and surrounding muscles (Behm & Anderson, 2006). Compared to stable training methods, unstable training creates greater stress on the neuromuscular system, and stress is essential for forcing the body to adapt to new stimuli (Selye, 1956). Meanwhile, the advantage of unstable training is that the unstable environment can better strengthen proprioception and enhance neuromuscular adaptation. The degree of muscle activation varies under different environmental conditions (Kraemer & Ratamess, 2004). When the external environment changes, proprioceptors in joints, tendons, and muscles become more sensitive to help the body perceive position and adjust posture (Proske & Gandevia, 2012). Unstable training is frequently applied in rehabilitation training (Pirauá et al., 2019; Lee et al., 2020; Behm & Colado, 2012), as it enhances sensory feedback, challenges the coordination of muscles and the nervous system, thereby improving body stability and control, which is crucial for promoting motor learning and neuroplasticity (Taube, Gruber & Gollhofer, 2008), particularly important for individuals with motor dysfunction. Additionally, unstable training is also widely used in sports performance enhancement, such as performing exercises in various positions on Swiss balls, Bosu balls, TRX suspension, balance boards and other unstable apparatuses. Among them, TRX suspension training is considered a good strategy to break the monotony of conventional exercises and change muscle activation according to training goals (Aguilera-Castells et al., 2020). TRX Suspension Training has two handles and straps in which users who are suspended from the straps use their body weight as resistance and perform multi-planar and multi-joint exercises (Gaedtke & Morat, 2015). However, the optimal load selection during unstable training remains controversial, which will be discussed in the subsequent review.

Muscle activation plays a crucial role in athletic performance, directly influencing an athlete’s strength, endurance, and movement efficiency, enabling them to optimize their performance in various sports activities (Tillin & Folland, 2014). Adequate muscle activation not only fully utilizes muscle strength and enhances the agility and responsiveness of the neuromuscular system but also reduces the risk of sports injuries, especially in movements involving multiple joints and complex actions, where the adequacy of muscle activation is even more critical (McHugh & Cosgrave, 2010; Enoka & Duchateau, 2017). Therefore, when conducting sports training, it is essential to pay close attention to the state of muscle activation, particularly during compound movements, where muscle activation is most crucial. Among these, a deep understanding of the differences in muscle activation effects among various movements is a crucial foundation and necessary prerequisite for formulating appropriate exercise prescriptions (Mausehund, Skard & Krosshaug, 2019), which helps to enhance training effectiveness and reduce injury risks, making sports training more scientific and targeted.

Previous studies on the activation of unstable cores have all held the attitude that they can change the activation effect. However, there are divergences in the activation of the upper and lower limbs. In the review by Batista et al. (2024), the effects of different surfaces on electromyographic activity were systematically evaluated, but the included studies showed high overall heterogeneity, few variables, and did not rigorously screen and group some key factors (such as training intensity, training time, subjects’ physical fitness, and training experience), making it difficult to clearly distinguish the effects of different unstable training. Moreover, the indicators for subgroup analysis were incomplete, lacking detailed analysis, and providing limited reference value. Based on this, this study aims to analyze the intervention effects of unstable training on core, upper limb and lower limb muscles activation, summarize more detailed muscle activation methods, and explore their intrinsic mechanisms, providing more targeted guidance for practical applications.

Methods

This study was performed in accordance with the Preferred Reporting Items for Systematic Reviews and Meta-Analyses (PRISMA) guidelines (Page et al., 2021). The review is registered with PROSPERO under the ID: CRD42024600670.

Search strategy

Relevant English and Chinese literature was searched in databases such as CNKI, VIP, Wanfang, PubMed, EBSCOhost, and Web of Science, with the search time ranging from the establishment of each database to the latest update time on November 26, 2024.

The search strategy is designed according to Boolean logic, and no medical subject terms (MeSH) of unstable training are found, so a comprehensive mix of free text terms is used to ensure thoroughness and accuracy: (“unstable” OR “instability” OR “surface”) AND (“biceps brachii” OR “trapezius” OR “serratus anterior” OR “triceps brachi” OR “biceps brachii” OR “trapezius” OR “anterior deltoid” OR “posterior deltoid” OR “latissimus dorsi” OR “rectus femoris” OR “vastus medialis” OR “vastus lateralis” OR “biceps femori” OR “gluteus medius” OR “soleus” OR “rectus abdominis” OR “internal oblique” OR “external oblique” OR “erector spinae”) AND (“muscle activation” OR “muscle activity”) AND (“electromyography” OR “EMG”).

Selection criteria

A set of a priori inclusion and exclusion criteria were used to evaluate study eligibility under the population, intervention, comparison, outcome, and study design (PICOS) framework. The inclusion criteria for this systematic review included human subjects of any gender, age, or training status as defined by McKay et al. (2022). Excluded if: (a) reviews, systematic reviews and meetings; (b) animal experiments or case studies; (c) repeated publication of literature; (d) no data can be obtained.

Studies were eligible for inclusion according to the following criteria: (a) Randomized cross-over design (RCD) published publicly; (b) the subjects were healthy people without chronic diseases; (c) the experimental group performs training on unstable surfaces (unstable training, UT), while the control group performs training on stable surfaces (stable training, ST); (d) studies that can provide clear sample sizes, means, and standard deviations; (e) EMG amplitude values of the agonist muscles were used as outcome indicators.

Data extraction

From the included studies, import the retrieved literature into Endnote X9 software to remove duplicates, two researchers (BZH and WS) initially evaluated the titles and abstracts and conducted a thorough review of the literature. Subsequently, a comprehensive review of the full texts was carried out. During this process, they independently extracted data from the selected studies. In case of any discrepancies, any differences regarding the inclusion of studies were resolved through consultation with a third researcher (LZY).

The extracted data includes: author, publication time, gender, sample size, age, training experience; author, practice method, practice intensity, exercise equipment, muscle activation (stabilizing surface effect, unstable surface effect, no effect difference), outcome indicators (rectus abdominis (RA), internal oblique (IO), external oblique (EO), erector spinae (ES), biceps brachii (BB), trapezius muscle (TM), deltoid muscle (DM), serratus anterior (SA), triceps brachii (TB), pectoralis major (PM), latissimus dorsi (LD), soleus muscle (SM), gluteus medius (GM), rectus femoris (RF), vastus medialis (VM), vastus lateralis (VL) and biceps femoris (BF)). Additionally, in the electromyography (EMG) analysis for evaluating muscle activation, the key variable involved is the root mean square (RMS) amplitude. The standardization process uses maximal voluntary isometric contraction (MVIC) and the maximal voluntary contraction (MVC) as the reference standards.

Assessment of methodological quality and statistical analyses

Quality evaluation of included literature was conducted using the Cochrane Risk of Bias Assessment Tool recommended by the Cochrane Handbook (Cochrane, 2019), classified as low risk (appropriate methods and measures were used, detailed descriptions were clear, the study design and implementation process was rigorous, and the results were reliable), high risk (randomization method is not explained in the study, which may lead to large bias in the study results and low confidence), or unclear risk (key information of the study is missing, unclear, or there is insufficient evidence to assess the risk of bias to determine whether the results of the study are reliable), and assessed for risk of bias in six aspects: selection bias, performance bias, measurement bias, follow-up bias, reporting bias, and other biases. The risk of bias in the included studies was assessed by two researchers (BZH and WS). The researchers conducted a comprehensive analysis of each assessment domain, and any discrepancies were resolved by consulting a third researcher (LZY).

Statistical analysis of the included data was performed using Stata 16.0 software, quality evaluation of the included literature was conducted through RevMan 5.3 software, and R was used for image processing. The standardized mean difference (SMD) between groups was calculated, and the 95% confidence interval (95% CI) was used for the comparison of studies. Among them, In comprehensive analyses of studies such as meta-analysis, the standardized mean difference (SMD) can standardize the effect sizes (such as the mean differences) of different studies, eliminating the influences of measurement units and sample sizes. This makes the results of different studies comparable and enables the evaluation of the overall effectiveness of the intervention measures across various studies, and if P < 0.05, it indicates statistical significance. I2 as a statistical measure to evaluate the heterogeneity among studies. When I2 < 25%, it indicates low heterogeneity; when 25% ≤ I2 ≤ 50%, it indicates moderate heterogeneity; and when I2 > 50%, it indicates high heterogeneity. Subgroup analysis, meta regression and sensitivity analysis were employed when the combined results exhibited high heterogeneity. The Egger method, Begg method, and funnel plot method were used to test for publication bias among the studies.

Results

Search results

A total of 1,971 relevant literature articles were retrieved in this study. Duplicate checking was performed using Endnote, and after removing duplicate articles, 1,681 remained. Based on inclusion and exclusion criteria, articles that clearly did not meet the standards were screened out by reading the titles and abstracts, resulting in 188 articles. After carefully reading the full texts, articles with inconsistent outcome indicators, missing data, or inconsistent reporting were excluded, ultimately yielding 28 articles (Fig. 1).

Figure 1 Flow chart of study selection.

Study characteristics

A total of 28 articles (Anderson et al., 2011; Bouillon et al., 2019; Calatayud et al., 2014; Czaprowski et al., 2014; Dunnick et al., 2015; Gündoğan, Aydin & Sağlam, 2023; Harris et al., 2017; Kohler, Flanagan & Whiting, 2010; Krause et al., 2018; Lawrence & Carlson, 2015; Lawrence et al., 2017; Lee, 2023; Lima et al., 2018; Liu, 2013; López-de-Celis et al., 2024; Luk et al., 2021; Norwood et al., 2007; Palma et al., 2021; Park & Yoo, 2013; Pirauá et al., 2014; Saeterbakken & Fimland, 2013; Snarr et al., 2016; Torres et al., 2017; Tucker et al., 2010; Uribe et al., 2010; Vera-Garcia, Grenier & McGill, 2000; Walker et al., 2023; Williams et al., 2020) were included in the study, all of which were randomized crossover trials. The total sample size of the included literature was 579 individuals, with the smallest sample size being 8 (Vera-Garcia, Grenier & McGill, 2000) and the largest being 43 (Luk et al., 2021). The average age ranged from 20 to 30 years, with no major diseases. All 28 studies indicated the gender of the participants, with 16 studies focusing on male participants (Calatayud et al., 2014; Dunnick et al., 2015; Gündoğan, Aydin & Sağlam, 2023; Lawrence & Carlson, 2015; Lawrence et al., 2017; Liu, 2013; Luk et al., 2021; Palma et al., 2021; Park & Yoo, 2013; Pirauá et al., 2014; Saeterbakken & Fimland, 2013; Torres et al., 2017; Uribe et al., 2010; Vera-Garcia, Grenier & McGill, 2000; Walker et al., 2023; Williams et al., 2020), 11 studies including both male and female participants (Anderson et al., 2011; Bouillon et al., 2019; Czaprowski et al., 2014; Harris et al., 2017; Kohler, Flanagan & Whiting, 2010; Krause et al., 2018; Lima et al., 2018; López-de-Celis et al., 2024; Norwood et al., 2007; Snarr et al., 2016; Tucker et al., 2010), and one study focusing on female participants (Lee, 2023). A total of 10 studies reported on training experience (prior experience in strength training) (Anderson et al., 2011; Calatayud et al., 2014; Dunnick et al., 2015; Lawrence & Carlson, 2015; Lawrence et al., 2017; Norwood et al., 2007; Pirauá et al., 2014; Saeterbakken & Fimland, 2013; Walker et al., 2023; Williams et al., 2020).

A total of 52 subcategories compared the differences in muscle activation between unstable and stable surfaces; in terms of exercise methods, 17 selected push-ups, six selected squats, one selected weighted half-squat, seven selected bench press, three selected shoulder press, one selected muscle-up, seven selected lunges, three selected bridge, one selected plank, one selected back extension, two selected Pilates, one selected isometric single-leg stand, one selected single-leg deadlift, and one selected sit-up. There were 31 subcategories with body weight as the exercise intensity; there were 21 studies with equipment load as the exercise intensity; in the dependent variables (intermediary media), six selected Swiss balls, one selected stable surface, one selected balance cone, nine selected Bosu balls, three selected Bosu balls and Swiss balls, one selected balance board and stability ball, 11 selected TRX suspension, two selected mats, four selected resistance bands, two selected Pilates equipment, four selected water tanks, two selected stepright, one selected Core coaster, one selected electric plate, three selected wobble boards, and one selected stability ball and stable surface for comparison; all 52 subcategories had multiple control groups, including the impact of different intermediary media on muscle activation. Nine studies indicated that muscle activation was effective on stable surfaces; 43 subcategories indicated that muscle activation was effective on unstable surfaces; 37 subcategories indicated no difference in muscle activation between stable and unstable surfaces (Table 1).

Table 1 The effect of muscle activation analysis.

Author/Year	Sample size/Gender	Age (M ± SD)	Training experience	Exercise method	Exercise intensity	Exercise equipment	Muscle activation	
Stable ↑	Unstable ↑	No difference	
Anderson et al. (2011)	15
(10 M, 5 F)	29.3 ± 6.4	10.4 ± 5.9	Push-up	Body-Weight	Stability ball★/Balance board※/Balance board※& Stability ball★	——	TB, RA, ES, IO,
SM	——	
Bouillon et al. (2019)	20
(7 M, 13 F)	23.4 ± 1.5	——	Lunge eccentric	Body-Weight	Stepright/Bosu ball	——	VM, RF, PL	TA, LG, BF	
Lunge concentric	Body-Weight	Stepright/Bosu ball	——	VM, RF, PL	TA, LG, BF	
Calatayud et al. (2014)	29 (M)	23.5 ± 3.1	>1 year	Push-up	Body-Weight	TRX suspension	——	TB, TM, AD, SM’, RA, ES,
QF	——	
Czaprowski et al. (2014)	33
(15 M, 18 F)	23.2 ± 2.5	——	Bridge	Body-Weight	Bosu ball, Swiss ball	——	——	RA, EO	
Dunnick et al. (2015)	20 (M)	24.1 ± 6.2	>1 year	Bench press	60% 1RM&
80% 1RM	Resistance band, kettlebell	——	LD, AD, MD	PM, TB	
Gündoğan, Aydin & Sağlam (2023)	14 (M)	20.4 ± 1.3	——	Squat	Body-Weight	Gymnastics mat/Bosu ball	——	VM, VL	——	
Harris et al. (2017)	25
(16 M, 9 F)	27.2 ± 4.0	——	Plank	Body-Weight	TRX suspension	——	OM	PM, RA, TM, DM	
Push-up	Body-Weight	TRX suspension	——	PM, RA, OM, TM, ES	DM, SA, OM	
Back up	Body-Weight	TRX suspension	——	DM, OM	SA, TM, RA	
Bridge	Body-Weight	TRX suspension	——	RA, ES	DM, OM,GM	
Kohler, Flanagan & Whiting (2010)	30
(24 M, 6 F)	30.0 ± 8.0	——	Shoulder press	75% 1RM	Swiss Ball	MD,TB	——	AD, TM	
Krause et al. (2018)	30
(15 M, 15 F)	23.9 ± 1.7	——	Lunges	Body-Weight	TRX suspension	——	HM, GM’, GM, LA	RF	
Lawrence & Carlson (2015)	15 (M)	24.2 ± 3.4	8.1 ± 4.3	Squat	60% 1RM	Elastic band, Barbell	——	EO, RA,
SM	RF, VL, VM, BF, ES	
Lawrence et al. (2017)	15 (M)	24.2 ± 2.7	9.9 ± 3.4	Bench press	60% 1RM&
75% 1RM	Resistance band, kettlebell	AD,PM,TB	BB	LD, MD,UT, PD	
Lee (2023)	26 (F)	22.1 ± 1.4	——	Pilates	Low	Pilates equipment	LD	IO, RA	——	
Heavy	Pilates equipment	——	GM’, IO, RA	——	
Lima et al. (2018)	19 (8 M, 11 F)	22.9 ± 4.1	——	Push-up	Body-Weight	TRX suspension	AD	BB	TB	
Liu (2013)	15 (M)	20.6 ± 1.7	——	Half squat	15% 1RM	Mat	——	RA, ES, QF, BF, TA	GM*	
López-de-Celis et al. (2024)	28
(18 M, 10 F)	24.5 ± 4.0	——	Isometric single-leg stand	10 kg	water tank	——	GM, VL, VM’	——	
Single-leg deadlift	10 kg	water tank	——	GM’, VM	VL	
Lateral lunges	10 kg	water tank	VM	——	GM’, VL	
Front rack forward lunges	10 kg	water tank	——	——	GM’, VL, VM	
Luk et al. (2021)	43 (M)	21.4 ± 1.8	——	Bridge	Body-Weight	TRX suspension	——	RA, RF, ES, BF, LM	——	
Norwood et al. (2007)	15
(10 M, 5 F)	29.3 ± 6.4	10.4 ± 5.9	Bench press	Original weight	Stabilization plane	——	——	LD, RA	
Bosu ball★	——	IO, BF	LD, SM	
Swiss ball※	——	IO, BF, SM	LD	
Bosu ball★, Swiss ball※	——	RA, IO, ES, BF, SM	LD	
Palma et al. (2021)	12 (M)	23.7 ± 3.0	——	Push-up	Body-Weight	Bosu ball/Balance disc	——	——	PM, EO, LM’, RA, LM	
TRX suspension	——	PM, EO, LM’	RA, LM	
Park & Yoo (2013)	14 (M)	21.4 ± 1.1	——	Push-up	Body-Weight	Swing board	——	TM, Lower SA, TB	PM, Middle SA	
Pirauá et al. (2014)	30 (M)	21.7 ± 2.5	>6 month	Push-up	Body-Weight	Swiss ball	——	TM	SA	
Saeterbakken & Fimland (2013)	15 (M)	23.3 ± 2.7	4.5 ± 1.8	Squat	Relative load	Electric plate	RF	SM	VM’, VL	
Bosu ball/Balance cone	RF	——	SM, VM’, VL	
Snarr et al. (2016)	20
(10 M, 10 F)	25.9 ± 5.6	——	PIKE Push-ups	Body-Weight	Bosu ball	——	RA, EO, RF	ES	
Swiss ball/TRX suspension/Core coaster	——	RA, EO, RF, ES	——	
Torres et al. (2017)	20 (M)	20.9 ± 1.8	——	Push-up	Body-Weight	Bosu ball	AD,PD	SA,
Middle &
Lower TM	PM, BB, TB, Upper TM	
Tucker et al. (2010)	30
(8 M, 22 F)	20.7 ± 3.2	——	Push-up	Body-Weight	Bosu ball	——	TM	SA	
Uribe et al. (2010)	16 (M)	24.2 ± 2.2	——	Bench press/Shoulder press	80% 1RM	Swiss Ball	——	AD	PM	
Vera-Garcia, Grenier & McGill (2000)	8 (M)	23.3 ± 4.3	——	Sit-ups	Body-Weight	Bosu ball, Swiss ball	——	RA, IO,	——	
Walker et al. (2023)	10 (M)	27.6 ± 7.9	3.5 ± 2.2	Muscle-up	Body-Weight	TRX suspension	——	Upper TM, BB, TB, FF	SA	
Williams et al. (2020)	12 (M)	25.3 ± 2.7	7.3 ± 2.4	Shoulder press	50% 1RM	Resistance band, kettlebell	——	BB, PM, RM, SA	——	
Note:

“——” indicates not described; ※ indicates upper limb instability; ★ indicates lower limb instability; ※ & ★ indicate both upper and lower limb instability; original weight is 9.1 kg; relative load is 20 kg with elastic band; RA, Rectus Abdominis, RF, Rectus Femoris; IO, Internal Oblique; EO, External Oblique; ES, Erector Spinae; BF, Biceps Femoris; LD, Latissimus Dorsi; PM, Pectoralis Major; LM, Lumbar Multifidus; LM’, Longissimus Muscle; GM’, Gluteus Medius; GM*, Gastrocnemius; SM, Soleus Muscle; SM’, Sternocleidomastoid Muscle; VL, Vastus Lateralis; VM, Vastus Medialis; TM, Trapezius; TB, Triceps Brachii; AT, Anterior Tibialis; SA, Serratus Anterior; AD, Anterior Deltoid; PL, peroneus longus; QF, quadriceps femoris; RM, Rhomboid Muscle; FF, Forearm Flexors; LG, lateral gastrocnemius.

Evaluation of the quality of the literature

The Cochrane Risk of Bias assessment tool was used to evaluate the methodology of the included literature. Among the 28 studies included in the analysis, 22 used outcome assessment blinding; 20 had complete outcome data; 11 were at low risk of bias, 10 at moderate risk of bias, and seven at high risk of bias (Fig. 2).

Figure 2 Bias risk assessment in the included literature.

Results of meta-analysis

The article conducted a meta-analysis on 28 included studies. The unstable surface was used as the experimental group, and the stable surface as the control group. The outcome indicators were divided into core muscles (rectus abdominis, internal oblique muscle, external oblique muscle, erector spinae), upper limb muscles (biceps brachii, trapezius, deltoid, serratus anterior, triceps brachii, pectoralis major, latissimus dorsi), and lower limb muscles (soleus, gluteus medius, rectus femoris, vastus medialis, vastus lateralis, biceps femoris).

Core

Rectus abdominis: The meta-analysis included 14 studies, with nine studies (21 effect sizes, 396 participants) comparing the effects of UT (experimental group) and ST (control group) on the rectus abdominis. Low heterogeneity (P = 0.14, I2 = 26%) was observed, and a fixed-effect model was used. The meta-analysis results showed a significant difference between the two groups (SMD = 0.32, 95% CI [0.18–0.46], P < 0.01) (Fig. 3). Sensitivity analysis showed that the combined results were stable.

Figure 3 Forest plot of core activation.

Studies: Anderson et al., 2011; Czaprowski et al., 2014; Harris et al., 2017; Kohler, Flanagan & Whiting, 2010; Lawrence & Carlson, 2015; Lee, 2023; Liu, 2013; Luk et al., 2021; Norwood et al., 2007; Palma et al., 2021; Saeterbakken & Fimland, 2013; Snarr et al., 2016; Vera-Garcia, Grenier & McGill, 2000.

Subgroup analysis results (Table 2) showed that UT had a better effect when the medium was a Bosu ball (SMD = 0.86 vs. 0.78); both exercise intensity and exercise method showed significant trends, with Body-Weight (SMD = 0.67) and sit-ups (SMD = 1.34 vs. 1.08) showing a more favorable trend.

Table 2 Subgroup analysis results of rectus abdominis.

Subgroup	Covariate	n	SMD	95% CI	P h	I 2	P d	
Exercise equipment	Swiss ball	99	0.67	[0.38–0.96]	<0.01	43%	0.81	
TRX	136	0.78	[0.53–1.02]	<0.01	0%		
Bosu ball	79	0.86	[0.53–1.20]	<0.01	35%		
Balance board	57	0.78	[0.40–1.17]	<0.01	0%		
Mat	30	0.51	[0.00–1.03]	0.05	0%		
Exercise intensity	Body-Weight	312	0.67	[0.50–0.83]	<0.01	15%	<0.01	
60% 1RM	75	−0.01	[−0.33 to 0.31]	0.94	0%		
Relative load	71	0.17	[−0.16 to 0.50]	0.32	0%		
Exercise method	Push-up	139	1.08	[0.82–1.33]	<0.01	12%	<0.01	
Bridge	144	0.40	[0.17–0.64]	<0.01	28%		
Squat	90	0.36	[0.07–0.66]	0.02	0%		
Shoulder press	105	−0.10	[−0.37 to 0.17]	0.46	0%		
Sit-ups	64	1.34	[0.95–1.74]	<0.01	0%		
Note:

n is the number of participants; I2 is the measure of heterogeneity between studies, expressed as a percentage; Ph is the P-value of the combined effect size; Pd is the P-value of the difference in effect size between subgroups.

Internal oblique muscle: A meta-analysis included 14 studies, of which six studies (14 effect sizes, 266 participants) compared the effects of UT (experimental group) and ST (control group) on the internal oblique. Low heterogeneity was observed (P = 0.03, I2 = 5%), and a fixed-effect model was used. The meta-analysis results showed a significant difference between the two groups (SMD = 0.38, 95% CI [0.20–0.56], P < 0.01) (Fig. 3). Sensitivity analysis showed that the combined results were stable.

Subgroup analysis results showed (Table 3) that UT had a better effect when the medium was a Swiss ball (SMD = 1.63 vs. 0.89); both exercise intensity and exercise method showed significant trends, with relative load (SMD = 1.39 vs. 0.52) and bench press (SMD = 1.78 vs. 0.74) showing a more favorable trend.

Table 3 Subgroup analysis results of internal oblique.

Subgroup	Covariate	n	SMD	95% CI	P h	I 2	P d	
Exercise equipment	Swiss ball	53	1.63	[1.18–2.08]	<0.01	0%	<0.01	
TRX	100	0.40	[0.12–0.68]	<0.01	0%		
Bosu ball	31	0.89	[0.36–1.43]	<0.01	26%		
Balance board	30	0.83	[0.29–1.36]	<0.01	45%		
Exercise intensity	Body-Weight	162	0.52	[0.30–0.74]	<0.01	0%	0.09	
Relative load	71	1.39	[0.48–2.30]	<0.01	82%		
Original weight	45	0.82	[−0.13 to 0.70]	0.18	0%		
Exercise method	Push-up	40	0.48	[0.03–0.93]	0.03	0%	<0.01	
Bridge	58	0.16	[−0.20 to 0.53]	0.39	2%		
Squat	45	0.29	[−0.13 to 0.70]	0.18	0%		
Bench press	45	1.78	[1.28–2.28]	<0.01	0%		
Sit-ups	32	0.74	[0.22–1.25]	<0.01	0%		
Note:

n is the number of participants; I2 is the measure of heterogeneity between studies, expressed as a percentage; Ph is the P-value of the combined effect size; Pd is the P-value of the difference in effect size between subgroups.

External oblique muscle: A meta-analysis included 14 studies, of which six studies (15 effect sizes, 249 participants) compared the effects of UT (experimental group) and ST (control group) on the external oblique. Moderate heterogeneity was observed (P = 0.26, I2 = 17%), and a fixed-effect model was used. The meta-analysis results showed a significant difference between the two groups (SMD = 0.38, 95% CI [0.20–0.56], P < 0.01) (Fig. 3). Sensitivity analysis showed that the combined results were stable.

Subgroup analysis results showed (Table 4) that UT had a better effect when the medium was a Swiss ball (SMD = 2.09 vs. 0.65); both exercise intensity and exercise method showed significant trends, with Body-Weight (SMD = 0.72) and sit-ups (SMD = 1.12 vs. 0.67) showing a more favorable trend.

Table 4 Subgroup analysis results of external oblique.

Subgroup	Covariate	n	SMD	95% CI	P h	I 2	P d	
Exercise equipment	Swiss ball	69	2.09	[1.66–2.51]	<0.01	0%	<0.01	
TRX	95	0.65	[0.35–0.94]	<0.01	43%		
Bosu ball	63	0.52	[0.16–0.88]	<0.01	17%		
Balance board	27	0.33	[−0.21 to 0.87]	<0.01	0%		
Exercise intensity	Body-Weight	163	0.72	[0.48, 0.96]	<0.01	8%	<0.01	
60% 1RM	105	−0.01	[−0.28 to 0.26]	0.93	0%		
Original weight	45	0.18	[−0.13 to 0.70]	0.18	0%		
Exercise method	Push-up	57	0.67	[0.29– 1.05]	<0.01	0%	<0.01	
Bridge	58	0.16	[−0.20 to 0.53]	0.38	2%		
Shoulder press	90	−0.07	[−0.37 to 0.22]	0.63	0%		
Squat	60	0.30	[−0.06 to 0.67]	0.18	0%		
Sit-ups	32	1.12	[0.57– 1.67]	<0.01	33%		
Note:

n is the number of participants; I2 is the measure of heterogeneity between studies, expressed as a percentage; Ph is the P-value of the combined effect size; Pd is the P-value of the difference in effect size between subgroups.

Erector spinae: A meta-analysis included 14 studies, with nine studies (18 effect sizes, 395 participants) comparing the effects of UT (experimental group) and ST (control group) on the erector spinae. There was high heterogeneity (P < 0.01, I2 = 87%), and a random-effects model was used. The meta-analysis results showed significant differences between the two groups (SMD = 0.60, 95% CI [0.17–1.02], P < 0.01) (Fig. 3). Sensitivity analysis showed stable combined results.

Subgroup analysis results showed (Table 5) that UT had better effects when the medium was the TRX suspension (SMD = 0.94); both exercise intensity and exercise mode showed significant trends, with Body-Weight (SMD = 0.94) and Bridge (SMD = 1.23 vs. 0.60) showing a more favorable trend. However, unexpectedly, the use of the Swiss ball (SMD = −0.41, P < 0.01), 60% 1RM (SMD = −0.36, P < 0.01), and Shoulder press (SMD = −0.36, P < 0.01) in the medium, exercise intensity, and exercise mode, respectively, had negative effects. Meanwhile, the subgroup analysis explained the high heterogeneity (I2 = 87%) in the erector spinae. Among the outcome indicators, there was no heterogeneity in outcome indicators (I2 = 0%), except for the exercise intensities of 60% 1RM and the exercise intensities of original weight, which indicates that exercise intensity is the main source of the heterogeneity in the erector spinae. To further examine the heterogeneity, a Meta-regression analysis was conducted (Table 6). The regression analysis indicated that the exercise intensity was indeed a source of heterogeneity (P < 0.05).

Table 5 Subgroup analysis results of erector spinae.

Subgroup	Covariate	n	SMD	95% CI	P h	I 2	P d	
Exercise equipment	Swiss ball	120	−0.41	[−0.66 to −0.15]	<0.01	0%	<0.01	
TRX	113	0.94	[0.66–1.21]	<0.01	0%		
Bosu ball	56	0.37	[−0.03 to 0.76]	0.07	0%		
Exercise intensity	Body-Weight	183	0.94	[0.72–1.16]	<0.01	0%	<0.01	
60% 1RM	105	−0.36	[−0.64 to −0.09]	<0.01	19%		
Original weight	45	0.16	[−0.26 to 0.57]	0.47	39%		
Relative load	45	−0.32	[−0.74 to 0.10]	0.13	0%		
Exercise method	Push-up	135	0.60	[0.36–0.85]	<0.01	0%	<0.01	
Bridge	111	1.23	[0.94–1.52]	<0.01	0%		
Shoulder press	90	−0.36	[−0.61 to −0.10]	<0.01	0%		
Squat	60	−0.20	[−0.56 to 0.16]	0.28	0%		
Note:

n is the number of participants; I2 is the measure of heterogeneity between studies, expressed as a percentage; Ph is the P-value of the combined effect size; Pd is the P-value of the difference in effect size between subgroups.

Table 6 Meta-regression of the training intensity of the erector spinae muscles.

ES	Coefficient std. err.	t	P >|t	[95% CI]	
Exercise intensity	−0.5331421	0.2356598	−2.26	0.036	[−1.026384 to −0.0399004]	
cons	1.556007	0.5179063	3.00	0.007	[0.4720171−2.639998]	

Upper limbs

Biceps brachii: A meta-analysis of four studies (seven effects, 101 participants) found that unstable training significantly increased biceps brachii activation compared to stable training (SMD = 0.52, 95% CI [0.23–0.80], P < 0.01), with moderate heterogeneity among studies (I2 = 26%) (Fig. 4). Sensitivity analysis showed stable combined results.

Figure 4 Forest plot of upper limbs activation (Part 1).

Studies: Dunnick et al., 2015; Kohler, Flanagan & Whiting, 2010; Lawrence et al., 2017; Lima et al., 2018; Park & Yoo, 2013; Pirauá et al., 2014; Torres et al., 2017; Tucker et al., 2010; Walker et al., 2023; Williams et al., 2020.

Subgroup analysis results (Table 7) showed that among the exercise method, ‘TRX Suspension’ and ‘Elastic Band & Kettlebell’ may be significant moderators (P < 0.01); among exercise intensities, ‘>50% 1RM’ and ‘Body-Weight’ are significant moderators (P < 0.01); among exercise methods, ‘Bench Press’ and ‘Muscle-Up’ may be significant moderators (P < 0.01); among exercise experience, ‘>1 year’ has a significant activation effect (P < 0.01).

Table 7 Subgroup analysis of meta-analysis results of biceps brachii.

Subgroup	Covariate	n	SMD	95% CI	P h	I 2	P d	
Exercise equipment	TRX	39	0.90	[0.43–1.38]	<0.01	10%	0.11	
Bosu ball	20	0.06	[−0.56 to 0.68]	0.84	–		
Elastic bands & kettlebells	54	0.53	[0.14–0.91]	<0.01	0%		
Exercise intensity	>50% 1RM	54	0.53	[0.14–0.91]	<0.01	0%	0.82	
Body-Weight	59	0.59	[0.22–0.97]	<0.01	55%		
Exercise method	Bench press	30	0.57	[0.05–1.09]	0.03	0%	0.15	
Push-up	39	0.31	[−0.14 to 0.76]	0.23	22%		
Muscle-up	20	1.27	[0.58–1.96]	<0.01	0%		
Shoulder press	24	0.48	[−0.11 to 1.06]	0.26	53%		
Exercise experience	>1 year	74	0.70	[0.37–1.04]	<0.01	13%	<0.01	
None	39	0.31	[−0.14 to 0.76]	0.18	22%		
Note:

n is the number of participants; I2 is the measure of heterogeneity between studies, expressed as a percentage; Ph is the P-value of the combined effect size; Pd is the P-value of the difference in effect size between subgroups.

Trapezius: A meta-analysis of eight studies (26 effects, 540 participants) found that unstable training significantly increased trapezius activation compared to stable training (SMD = 0.23, 95% CI [0.12–0.35], P < 0.01), with no heterogeneity among studies (I2 = 0%) (Fig. 4). Sensitivity analysis showed stable combined results.

Subgroup analysis results (Table 8) showed that among the exercise method, ‘Bosu Ball’ and ‘Wobble Board’ may be significant moderators (P < 0.01); among exercise intensities, ‘Body-Weight’ may be a significant moderator (P < 0.01); among exercise methods, ‘Push-Up’ may be a significant moderator (P < 0.01); among exercise experience, ‘none’ may be a significant moderator (P < 0.01); among contraction modes, ‘concentric contraction’ had a better activation effect (P = 0.05).

Table 8 Subgroup analysis of meta-analysis results of trapezius muscle.

Subgroup	Covariate	n	SMD	95% CI	P h	I 2	P d	
Exercise equipment	TRX	40	0.11	[−0.33 to 0.55]	0.62	0%	0.32	
Swiss ball	150	0.11	[−0.11 to 0.33]	0.32	0%		
Bosu ball	240	0.27	[0.09–0.46]	<0.01	9%		
Swing board	56	0.57	[0.19–0.95]	<0.01	0%		
Elastic bands & kettlebells	54	0.18	[−0.20 to 0.56]	0.36	0%		
Exercise intensity	>60% 1RM	120	0.00	[−0.25 to 0.26]	0.98	0%	0.11	
50% 1RM	24	0.17	[−0.40 to 0.73]	0.56	0%		
Body-Weight	396	0.31	[0.17–0.45]	<0.01	0%		
Exercise method	Bench press	30	0.19	[−0.32 to 0.69]	0.47	0%	0.04	
Push-up	356	0.33	[0.18–0.48]	<0.01	0%		
Shoulder press	114	−0.01	[−0.27 to 0.25]	0.93	0%		
Muscle-up	40	0.11	[−0.33 to 0.55]	0.62	0%		
Exercise experience	>6 months	244	0.13	[−0.05 to 0.30]	0.16	0%	0.1	
None	296	0.33	[0.17–0.49]	<0.01	0%		
Contraction mode	Eccentric	48	0.34	[−0.07 to 0.75]	0.10	0%	0.83	
Concentric	48	0.40	[0.00–0.81]	0.05	0%		
Note:

n is the number of participants; I2 is the measure of heterogeneity between studies, expressed as a percentage; Ph is the P-value of the combined effect size; Pd is the P-value of the difference in effect size between subgroups.

Deltoid: A total of seven studies (17 effects, 327 participants) were included. Meta-analysis found that unstable training compared to stable training did not significantly improve deltoid activation (SMD = −0.06, 95% CI [−0.21 to 0.09], P = 0.44), and there was no heterogeneity among studies (I2 = 0%) (Fig. 4). Sensitivity analysis showed that the combined results were stable.

Subgroup analysis results showed (Table 9) that no significant effects were observed for exercise method (P = 0.41, I2 = 0%), exercise intensity (P = 0.44, I2 = 0%), exercise mode (P = 0.44, I2 = 0%), training experience (P = 0.44, I2 = 0%), and contraction mode (P = 0.54, I2 = 0%) on deltoid muscle activation (P > 0.05). However, in the exercise mode, ‘push-ups’ were unfavorable for deltoid muscle activation (P = 0.03, 95% CI [−0.94 to −0.04], I2 = 34%); although the Bosu ball showed a negative effect (95% CI [−1.42 to −0.13], P < 0.05), since only one study was included, the reliability was low, and thus it was not included for reference.

Table 9 Subgroup analysis results of the deltoid.

Subgroup	Covariate	n	SMD	95% CI	P h	I 2	P d	
Exercise equipment	Swiss ball	154	−0.01	[−0.23 to 0.22]	0.96	0%	0.15	
TRX	19	−0.21	[−0.85 to 0.43]	0.52	–		
Bosu ball	20	−0.78	[−1.42 to −0.13]	0.02	–		
Elastic bands & kettlebells	174	−0.02	[−0.23 to 0.19]	0.87	0%		
Exercise intensity	60% 1RM &
80% 1RM	80	0.12	[−0.19 to 0.43]	0.43	0%	0.44	
60% 1RM &
75% 1RM	120	−0.12	[−0.38 to 0.13]	0.33	0%		
Body-Weight	103	−0.18	[−0.45 to 0.10]	0.21	0%		
50% 1RM	24	0.13	[−0.43 to 0.70]	0.65	0%		
Exercise method	Bench press	142	−0.03	[−0.26 to 0.21]	0.83	0%	0.14	
Push-up	39	−0.49	[−0.94 to −0.04]	0.03	34%		
Shoulder press	146	0.02	[−0.21 to 0.25]	0.89	0%		
Exercise experience	>1 year	224	−0.01	[−0.19 to 0.18]	0.93	0%	0.32	
None	103	−0.18	[−0.45 to 0.10]	0.21	0%		
Contraction mode	Eccentric	72	0.15	[−0.18 to 0.48]	0.37	0%	0.52	
Concentric	72	0.00	[−0.33 to 0.32]	0.99	0%		
Note:

n is the number of participants; I2 is the measure of heterogeneity between studies, expressed as a percentage; Ph is the P-value of the combined effect size; Pd is the P-value of the difference in effect size between subgroups.

Serratus anterior: A total of four studies (nine effects, 120 participants) were included. Meta-analysis found no heterogeneity among studies (I2 = 0%), and a fixed-effect model was used for analysis. Further analysis found that unstable training could significantly enhance serratus anterior activation (SMD = 0.33, 95% CI [0.07–0.59], P = 0.01), and there was no heterogeneity among studies (I2 = 0%) (Fig. 4). Sensitivity analysis showed that the combined results were stable.

Subgroup analysis results (Table 10) showed that no significant activation effects were observed on the serratus anterior muscle activation with respect to exercise method, exercise intensity, exercise mode, exercise experience, or contraction mode (P > 0.05). However, UT showed a negative effect in studies where the exercise method was a Swiss ball (95% CI [−0.82 to −0.09], P < 0.05).

Table 10 Subgroup analysis results of the serratus anterior.

Subgroup	Covariate	n	SMD	95% CI	P h	I 2	P d	
Exercise equipment	TRX	39	1.36	[−0.31 to 3.03]	0.11	90%	0.03	
Swing board	56	0.22	[−0.15 to 0.59]	0.25	0%		
Swiss ball	60	−0.46	[−0.82 to −0.09]	0.01	0%		
Bosu ball	80	−0.02	[−0.56 to 0.53]	0.95	66%		
Elastic bands & kettlebells	24	0.22	[−0.34 to 0.79]	0.44	0%		
Exercise intensity	Body-Weight	235	−0.05	[−0.32 to 0.22]	0.72	52%	0.39	
50% 1RM	24	0.22	[−0.34 to 0.79]	0.44	0%		
Exercise method	Push-up	215	−0.17	[−0.36 to 0.02]	0.09	53%	0.08	
Muscle-up	20	0.49	[−0.14 to 1.12]	0.13	0%		
Shoulder Press	24	0.22	[−0.34 to 0.79]	0.44	0%		
Exercise experience	>6 months	104	−0.03	[−0.40 to 0.35]	0.90	42%	0.86	
None	155	−0.02	[−0.36 to 0.32]	0.92	55%		
Contraction mode	Eccentric	38	0.22	[−0.23 to 0.67]	0.34	0%		
Concentric	38	0.36	[−0.09 to 0.82]	0.12	0%		
Note:

n is the number of participants; I2 is the measure of heterogeneity between studies, expressed as a percentage; Ph is the P-value of the combined effect size; Pd is the P-value of the difference in effect size between subgroups.

Triceps brachii: Six studies (12 effects, 191 participants) were included. Meta-analysis revealed no heterogeneity among studies (I2 = 0%), and a fixed-effect model was used for analysis. Further analysis found that unstable training significantly increased triceps brachii activation (SMD = 0.24, 95% CI [0.04–0.45], P = 0.02) (Fig. 5). Sensitivity analysis showed that the combined results were stable.

Figure 5 Forest plot of upper limbs activation (Part 2).

Studies: Dunnick et al., 2015; Lawrence et al., 2017; Lima et al., 2018; Palma et al., 2021; Park & Yoo, 2013; Torres et al., 2017; Uribe et al., 2010; Walker et al., 2023; Williams et al., 2020.

Subgroup analysis results show (Table 11) that in the exercise method, the ‘Swiss Ball’ (P < 0.01, 95% CI [−1.45 to −0.32], I2 = 70%), in exercise intensity, ‘60% 1RM & 75% 1RM’ (P < 0.01, 95% CI [−1.12 to −0.28], I2 = 58%), and in exercise experience, ‘>1 year’ (P = 0.02, 95% CI [−0.40 to −0.04], I2 = 68%) significantly reduced triceps brachii muscle activation. Meanwhile, in exercise intensity, ‘Body-Weight’ (P = 0.04, 95% CI [0.01–0.61], I2 = 0%) had a significant effect. The article did not observe a significant impact of exercise method on triceps brachii muscle activation (P > 0.05). In the contraction mode, there was also not observe a significant effect (P > 0.05).

Table 11 Subgroup analysis results of the triceps brachii.

Subgroup	Covariate	n	SMD	95% CI	P h	I 2	P d	
Exercise equipment	TRX	39	0.26	[−0.19 to 0.71]	0.25	0%	<0.01	
Swiss ball	90	−0.88	[−1.45 to −0.32]	<0.01	70%		
Swing board	28	0.32	[−0.20 to 0.85]	0.23	0%		
Bosu ball	20	0.40	[−0.23 to 1.03]	0.21	–		
Elastic bands & kettlebells	104	0.19	[−0.08 to 0.46]	0.18	0%		
Exercise intensity	60% 1RM & 80% 1RM	80	0.20	[−0.11 to 0.51]	0.20	58%	<0.01	
60% 1RM & 75% 1RM	120	−0.70	[−1.12 to −0.28]	<0.01	0%		
50% 1RM	24	0.14	[−0.43 to 0.70]	0.64	0%		
Body-Weight	87	0.31	[0.01–0.61]	0.04	64%		
Exercise method	Bench press	110	0.06	[−0.21 to 0.32]	0.68	0%	0.07	
Push-up	67	0.28	[−0.06 to 0.62]	0.11	0%		
Shoulder press	114	−0.53	[−1.09 to 0.02]	0.06	75%		
Muscle-up	20	0.43	[−0.20 to 1.06]	0.18	0%		
Exercise experience	>1 year	244	−0.22	[−0.40 to −0.04]	0.02	68%	0.01	
None	67	0.28	[−0.06 to 0.62]	0.11	0%		
Contraction mode	Eccentric	64	0.34	[−0.01 to 0.69]	0.06	0%	0.54	
Concentric	64	0.19	[−0.16 to 0.54]	0.29	0%		
Note:

n is the number of participants; I2 is the measure of heterogeneity between studies, expressed as a percentage; Ph is the P-value of the combined effect size; Pd is the P-value of the difference in effect size between subgroups.

Pectoralis major: Currently, the included original trials (18 effects, 302 participants) consistently found that unstable training compared to stable training had no significant effect on pectoralis major activation (Fig. 5).

Latissimus dorsi: Currently, the included original trials (10 effects, 154 participants) consistently found that unstable training compared to stable training had no significant effect on latissimus dorsi activation (Fig. 5).

Lower limbs

Soleus: Four studies (11 effects, 330 participants) were included. Meta-analysis found that unstable training significantly increased the activation of the soleus compared to stable training (SMD = 0.65, 95% CI [0.42–0.87], P < 0.01), with moderate heterogeneity between studies (I2 = 48%) (Fig. 6). Sensitivity analysis showed that the combined results were stable.

Figure 6 Forest plot of lower limbs activation (Part 1).

Studies: Anderson et al., 2011; Krause et al., 2018; Lawrence & Carlson, 2015; López-de-Celis et al., 2024; Norwood et al., 2007; Saeterbakken & Fimland, 2013.

Subgroup analysis results (Table 12) showed that UT had a statistically significant effect in the squat exercise mode (SMD = 0.38).

Table 12 Subgroup analysis results of soleus.

Subgroup	Covariate	n	SMD	95% CI	P h	I 2	P d	
Exercise method	Squat	75	0.38	[0.06–0.71]	0.02	0%	<0.01	
Note:

n is the number of participants; I2 is the measure of heterogeneity between studies, expressed as a percentage; Ph is the P-value of the combined effect size; Pd is the P-value of the difference in effect size between subgroups.

Gluteus medius: Two primary studies (five effects, 284 participants) were included. Meta-analysis found that unstable training significantly increased the activation of the gluteus medius compared to stable training (SMD = 0.28, 95% CI [0.05–0.52], P = 0.02), with no heterogeneity between studies (I2 = 0%) (Fig. 6). Sensitivity analysis showed that the combined results were stable.

Subgroup analysis results (Table 13) showed that in terms of exercise mode and intensity, ‘lunges’ (SMD = 0.21, 95% CI [−0.22 to 0.65], P = 0.33) and ‘10 kg’ (SMD = 0.19, 95% CI [−0.08 to 0.45], P = 0.16) were not significant moderating factors, and they did not significantly increase the activation of the gluteus medius.

Table 13 Subgroup analysis results of gluteus medius.

Subgroup	Covariate	n	SMD	95% CI	P h	I 2	P d	
Exercise method	Lunges	86	0.21	[−0.22 to 0.65]	0.33	52%	0.33	
Exercise intensity	10 kg	112	0.19	[−0.08 to 0.45]	0.16	0%	0.19	
Note:

n is the number of participants; I2 is the measure of heterogeneity between studies, expressed as a percentage; Ph is the P-value of the combined effect size; Pd is the P-value of the difference in effect size between subgroups.

Rectus femoris: Four studies (10 effects, 370 participants) consistently found that unstable training compared to stable training had no significant effect on the activation of the rectus femoris (Fig. 7).

Figure 7 Forest plot of lower limbs activation (Part 2).

Studies: Bouillon et al., 2019; Gündoğan, Aydin & Sağlam, 2023; Krause et al., 2018; Lawrence & Carlson, 2015; López-de-Celis et al., 2024; Saeterbakken & Fimland, 2013.

Subgroup analysis results show (Table 14) that in terms of exercise method and medium, both ‘squat’ (SMD = −0.34, 95% CI [−0.66 to −0.01], P = 0.04) and ‘Bosu ball’ (SMD = −0.49, 95% CI [−0.87 to −0.11], P = 0.01) had significant moderating factors, but had a negative activation effect on the rectus femoris.

Table 14 Subgroup analysis results of rectus femoris.

Subgroup	Covariate	n	SMD	95% CI	P h	I 2	P d	
Exercise method	Lunges	110	−0.09	[−0.36 to 0.17]	0.50	0%	0.25	
Squat	75	−0.34	[−0.66 to −0.01]	0.04	48%		
Exercise equipment	Bosu ball	55	−0.49	[−0.87 to −0.11]	0.01	0%	0.01	
Note:

n is the number of participants; I2 is the measure of heterogeneity between studies, expressed as a percentage; Ph is the P-value of the combined effect size; Pd is the P-value of the difference in effect size between subgroups.

Vastus lateralis: Three studies (nine effects, 370 participants) consistently found that unstable training had no significant effect on the activation of the vastus lateralis compared to stable training (Fig. 7).

Vastus medialis: Four studies (13 effects, 530 participants) consistently found that unstable training had no significant effect on the activation of the vastus medialis compared to stable training (Fig. 7).

Biceps femoris: Four studies (12 effects, 400 participants) consistently found that unstable training had no significant effect on the activation of the biceps femoris compared to stable training (Fig. 7).

Publication bias analysis results

Among the included studies, the effect on core muscle activation was the most significant, so it was used as an example for funnel plot analysis (Fig. 8). The funnel plot showed that the distribution of literature was roughly symmetrical, indicating no publication bias or other biases.

Figure 8 Publishing bias funnel plots.

Discussion

A total of 28 studies (579 participants) were included to investigate the effects of unstable training on muscle activation, with stable training used for comparative analysis. The meta-analysis results indicate that compared to training on stable surfaces, unstable training significantly enhances muscle activation in core muscles (rectus abdominis, internal oblique, external oblique, and erector spinae), upper limb muscles (biceps brachii, trapezius, serratus anterior, and triceps brachii), and lower limb muscles (soleus and gluteus medius). Subgroup analysis shows that participant characteristics (with or without training experience) and muscle activation protocols (exercise type, intensity, medium, and contraction mode) potentially moderate the combined effects of unstable training on various muscle indicators. The results of each outcome indicator will be discussed and analyzed in depth below.

Core

The core muscles play a bridging role in the transfer of strength between the upper and lower extremities (Kibler, Press & Sciascia, 2006). Compared to stable environments, unstable environments require continuous balance adjustments, and the core muscles must participate more actively (Behm & Colado, 2012; Luo et al., 2022). The aggregated results show that unstable training can significantly increase the activation of core muscles (P < 0.01). The overall findings support the previous meta-analysis by Batista et al. (2024), which found that unstable training significantly enhances the activation of core muscles (rectus abdominis, external oblique, internal oblique, erector spinae, and lumbar multifidus). However, Batista et al. (2024) included experiments with large variations in study design, participant characteristics (such as age, gender, training level), and training variables (such as frequency, duration, intensity), which may lead to high overall heterogeneity in the included studies. Additionally, key factors such as training intensity, training duration, participants’ physical fitness, and training experience were not strictly screened and grouped, making it difficult to clearly distinguish the effects of different unstable exercises, which may affect the comprehensive understanding of core activation. Furthermore, fewer studies were included in the research on the internal oblique and lumbar multifidus (two and four studies, respectively), which may affect the statistical significance and accuracy of effect size estimation, making the results more susceptible to random factors, leading to instability and a higher risk of bias. Sensitivity and detailed subgroup analyses were not conducted. In contrast, the article included at least six studies on core muscles, providing a detailed analysis of different unstable equipment, training intensity, and exercise methods, with higher quality of included literature.

Meanwhile, the subgroup analysis of the article further indicates that, compared to stable training, exercise equipment, practice intensity, and practice method are significant moderating factors affecting core muscle activation. Among them, using the Bosu ball, body weight, and sit-ups are the best schemes for activating the RA. This is consistent with the results of the systematic review by Oliva-Lozano & Muyor (2020), which indicated that the maximum activation effect in RA, EO, and ES is found on body weight, and unstable sit-ups on the Bosu ball achieved the highest RA activation effect. This is because sit-ups fix the angles of the upper and lower limbs, avoiding the compensation of the lumbar or hip muscle groups, thus enabling the rectus abdominis (RA) to contract in isolation (Crommert et al., 2021). The main functions of the RA are spinal flexion and the maintenance of intra-abdominal pressure. Compared with the multi-muscle group synergistic effect in weight-bearing training (especially the compensation of the lumbar muscle groups), participants with insufficient strength or those lacking in experience and skill levels may rely on inertia to complete the movements. However, due to the lever advantage of no load in bodyweight training, the influence of confounding factors is relatively small. Therefore, it is easier to activate the RA (Maeo et al., 2013; Saeterbakken et al., 2014). Moreover, in terms of IO activation, using the Swiss ball, relative load, and bench press are the best schemes for activating IO. This may be because the IO is located deep within the external oblique (Flament, 2006), and the bench press, as a compound strength training movement, involves the collaboration of multiple joints and muscle groups. When the bench press is performed in an unstable environment, the increase in weight enhances neural activation (Nuzzo et al., 2017), and to maintain body balance, the internal oblique muscle is more involved in spinal stability (Maeo et al., 2013). Bezerra et al. (2020) also found that using the Swiss ball for upper-body instability yielded more significant effects than lower-body instability. Using the Swiss ball, body weight, and sit-ups are the best schemes for activating the EO. This may be related to the external oblique’s primary role in trunk rotation, lateral flexion, and assisting in maintaining trunk stability (Murofushi et al., 2023), and all the included studies involved upper-body instability. Using the TRX suspension, body weight, and bridge are the best schemes for activating the ES. This finding is also consistent with previous research by Lee, Park & Lee (2015). Since the muscle activity of the ES during the bridge at a high suspension position was significantly higher than at a low suspension position (P < 0.05). Additionally, studies by Yoon et al. (2018) and Bi et al. (2012) also confirmed significant effects of the erector spinae during unstable bridges. However, unexpectedly, the use of the Swiss ball, 60% 1RM, and shoulder press resulted in a significant reduction in ES activation. Specifically, the primary task of the erector spinae is to maintain an upright posture along the spine (Desmons et al., 2024), and compared to shoulder press performed on a stable surface, shoulder press on a Swiss ball rely more on shoulder, chest, and core stability, while the significant reduction in force production may be due to the muscles around the shoulder complex prioritizing stability over force generation (Marquina et al., 2021), which leads to more trunk control required on unstable surfaces, with the activation of the abdominal and lateral abdominal muscles often exceeding that of the erector spinae, which is relatively inhibited in this process. However, in these situations, unstable conditions produce better results than stable conditions. For instance, when the prime movers become fatigued due to a large number of repetitions and a long duration of exercise, the erector spinae muscles compensate to ensure the execution of technical movements and maintain balance (Campbell et al., 2014). Compared with those who have rich exercise experience, individuals with less exercise experience tend to have a higher level of muscle activation. This is because those with lower experience have poorer neuromuscular adaptability (Chulvi-Medrano et al., 2010) and are unable to control the muscles well enough to complete the corresponding technical movements. As a result, the compensatory effect of the erector spinae muscles increases in order to maintain body stability. At the same time, experienced individuals can reduce the need for compensation from the erector spinae muscles by synchronizing their breathing with core contractions. Conversely, those with less experience are unable to do so (Huxel Bliven & Anderson, 2013).

In summary, unstable training can effectively activate the abdominal core muscles. For rectus abdominis, Bosu balls, body weight, and sit-ups are effective. Swiss balls, relative load, and bench press work well for internal oblique. Swiss balls, body weight, and sit-ups are suitable for external oblique, while TRX suspension trainers, body weight, and side planks are recommended for erector spinae. Targeted core stability training is best performed simultaneously with core activation to have marginal benefits for sports performance (Reed et al., 2012).

Upper limbs

Properly activating the upper limb muscles not only enhances strength output but also improves neuromuscular coordination and reduces the risk of injury (Nekar et al., 2024). Particularly during high-intensity training, pre-training upper limb activation helps achieve better muscle coordination in subsequent training, improves training effectiveness, and reduces technical errors caused by fatigue (Aagaard et al., 2002). The article finds that unstable training can increase the activation levels of the biceps brachii, trapezius, serratus anterior, and triceps brachii. Subgroup analysis results indicate that individuals with more than 1 year of exercise experience achieve the best activation effects by performing the muscle-up exercise using a TRX suspension combined with body weight during BB training. This may be because exercising on a TRX suspension requires greater synergistic effort from the upper limbs and core muscles, and suspension exercises can achieve training effects close to traditional 70% 1RM intensity, making them more suitable for advanced training by highly skilled individuals (Vural et al., 2023). Meanwhile, in the systematic review by Aguilera-Castells et al. (2020), it was pointed out that with the variation of the gripping methods of the TRX suspension trainer, changes in the trunk-leg inclination angle and hip flexion angle can lead to significant differences in the activation effect of the biceps brachii. Using a supinated grip or a neutral grip can increase the activation level of the biceps brachii (Vural et al., 2023). For individuals with limited training experience, the most effective method for TM activation is the push-up on a Bosu ball combined with body weight, with more pronounced effects during the concentric phase. This may be because the Bosu ball provides greater instability (Gruber & Gollhofer, 2004), requiring higher control of the scapula on an unstable surface to prevent scapular displacement under unstable conditions (Anderson & Behm, 2005; Saeterbakken, van den Tillaar & Fimland, 2011). From a physiological perspective, under unstable conditions, concentric contractions require higher muscle activation to counteract the dual influence of gravity and instability. This contraction increases the recruitment of muscle fibers, particularly in the upward and medial scapular movements, where the upper and middle trapezius are more highly activated to maintain shoulder stability (Araújo et al., 2011). Additionally, concentric contractions under unstable conditions require rapid responses to counteract minor displacements caused by instability. These rapid responses help increase the number of muscle units recruited and enhance the muscle’s instantaneous control ability (Seo et al., 2013). SA should avoid using the Swiss ball during exercises to reduce the activation of the serratus anterior. Cools et al. (2007) found that push-ups on a Swiss ball may result in insufficient upward rotation of the scapula or excessive involvement of the upper trapezius, leading to decreased activation of the serratus anterior. This is related to the higher center of gravity and greater instability of the Swiss ball (Yue, Gong & Zhou, 2015). Although this benefits the activation of core muscles, the high instability of the ball may cause the body to prioritize the activation of other muscle groups to maintain stability, resulting in relatively reduced activation of the serratus anterior (Norwood et al., 2007). However, when performing push-ups on a Swiss ball, if the method of increasing the trunk inclination angle is adopted, the muscle activation effect of the serratus anterior will be somewhat higher. This is because as the trunk inclination angle increases, the SA needs to contract more actively to resist the scapular retraction caused by gravity (Freeman et al., 2006). More control over the scapula is the key, but correspondingly, the difficulty of the movement will be greater, and its practical value is not very high. Finally, for TB training, bodyweight training yields the most significant activation effects, while individuals with more than 1 year of training experience achieve greater activation on a Swiss ball with intensities exceeding 60% of 1RM. Training with Swiss balls may lead to negative effects. This could be because combined strength training centered around Swiss balls can improve body stability, reduce the degree of shift in body center of gravity (Liu, 2012), resulting in more attention and energy being diverted to the involvement of core and stabilizing muscles, forcing the core muscles to collaborate more to maintain balance, thereby reducing the activation of the triceps brachii (Anderson & Behm, 2005). Behm & Anderson (2006) found that under unstable conditions, due to increased joint stiffness during movement execution, coordination, force, and performance may be hindered. Movements may become harder to control, leading to insufficient activation of the target muscles (McBride, Cormie & Deane, 2006). Experienced exercisers are better at recruiting target muscles on stable surfaces. Kowalski et al. (2022) discovered that under unstable conditions, experienced individuals tend to engage more synergistic muscles to maintain balance, thereby reducing direct involvement of the target muscles, while beginners may rely more on the target muscles for movement execution due to their lower coordination and synergistic muscle involvement. However, bodyweight exercises significantly enhance training intensity. In resistance training, the triceps brachii primarily functions in elbow extension (Calatayud et al., 2016). Compared to training in stable environments, unstable training primarily provides additional engagement to maintain elbow stability during triceps activation (Lehman et al., 2006). Additionally, when the glenohumeral joint is in an unstable environment, triceps activation increases (De Mey et al., 2014). During bodyweight training, the load is relatively light, increasing reliance on smaller muscles like the triceps, which are more activated to assist in controlling instability (Carroll, Riek & Carson, 2001). Conversely, in high-load training under unstable conditions, due to the relatively large weight, the metabolic pressure index (blood lactate concentration) of the triceps brachii is lower than that under stable conditions. This indicates insufficient mechanical tension. Due to neural inhibition and compensatory patterns, TB fails to effectively accumulate metabolic pressure, weakens the muscle hypertrophy signaling pathway, and results in a lower activation effect (Maté-Muñoz et al., 2014).

In conclusion, unstable training can promote greater neuromuscular adaptations, such as reduced co-contraction, improved coordination, and increased confidence in performing skills (Behm & Anderson, 2006). Trainers should develop a feasible training plan based on their own training experience, level, and goals, and implement it during the training process. Since upper limb unstable training is quite challenging, it is recommended that trainers with insufficient training experience can choose to perform push-up exercises using a Bosu ball in combination with body weight to optimize the activation effect. On the other hand, trainers or athletes with more experience should avoid using a Swiss ball for activation during serratus anterior exercises.

Lower limbs

Activation of lower limb muscles is not only controlled by local motor functions but also neuro-coupled with upper limb movements, influencing overall motor coordination. During maximal effort movements, activating lower limb muscles can regulate whole-body force output and improve athletic performance (Huang & Ferris, 2009). The study found that unstable training significantly activates the soleus and gluteus medius muscles, and subgroup analysis results indicate that squats have a significant effect on soleus muscle activation. The soleus muscle is located on the posterior side of the calf, deep to the gastrocnemius muscle (Standring, 2016). The soleus muscle is one of the key muscles in the lower limb for maintaining posture and balance, playing a critical role in posture control (Kouzaki & Masani, 2012). Under unstable conditions, the body requires more complex neuromuscular coordination to maintain balance, which improves neuromuscular coordination, increases the H-reflex amplitude of the soleus muscle, and may indicate disinhibition of the neuromuscular pathway and higher adaptability (Friesenbichler, Lepers & Maffiuletti, 2015). Deep muscles are additionally activated to meet posture control demands in optimizing athletic performance (Hibbs et al., 2008). Unstable environments significantly increase the demand for posture control, enhancing the soleus muscle’s ability to respond to postural disturbances, allowing it to activate more quickly during dynamic tasks to maintain ankle stability and body balance (Lawrence & Carlson, 2015). Behm & Anderson (2006) also indicated that during squats, the soleus muscle may bear more postural responsibility than the quadriceps, making it logical to expect greater soleus muscle activity under unstable conditions. Additionally, it is noteworthy that in soleus muscle activation, bench press and push-ups could not be further analyzed due to insufficient study numbers, but based on current analysis, they still show significance (Anderson et al., 2011; Norwood et al., 2007). In gluteus medius muscle activation, although unstable training has an activating effect on the GM (P = 0.02), it did not show statistical significance in subgroup analysis. Krause et al. (2009) have fully demonstrated that the gluteus medius exhibits a greater muscle activation trend on surface instability. Youdas et al. (2015) indicates that unstable surface training provides a stronger stimulus for gluteus medius activation compared to stable surfaces, a result of increased load on the hip abductors. The abductor load forces the body to stabilize the pelvis and maintain balance by activating the gluteus medius, which must bear more load to prevent excessive pelvic tilt and maintain hip stability, significantly increasing its demand for stabilization function (Krause et al., 2009). Additionally, the hip abductor load enhances the nervous system’s recruitment frequency and intensity of the gluteus medius to coordinate hip and pelvic stability, resulting in a significant increase in its electromyographic activity, showing higher muscle activation levels compared to stable surfaces (Bishop et al., 2018). Compared to movements primarily involving hip activities, the lunge action mainly involves hip flexion and knee coordination, with limited activation demand for the gluteus medius as the primary hip abductor muscle (DiStefano et al., 2009). However, it remains unclear whether and how hip stabilizing muscles adjust their activation patterns from stable to unstable support conditions during weighted leg extension tasks (Kibele, 2017). In terms of training experience, untrained individuals have a much lower activation effect in unstable training compared to those with experience. This is because experienced individuals can pre-activate and mobilize the gluteus medius muscle before the movement starts to reduce the amplitude of pelvic sway (Snyder et al., 2009). However, it should be noted that it is necessary to avoid incorrect movements that force the tensor fasciae latae muscle to be overly involved, which may inhibit the neural drive of the gluteus medius muscle and increase the pressure on the patellofemoral joint (Bolgla et al., 2008). The RF, being the most superficial muscle of the quadriceps, exhibits different oxygen delivery strategies and motor responses compared to deeper muscles during movement (Koga et al., 2017). In unstable training, there is often a greater emphasis on the coordinated work of core and postural muscles rather than isolated activation of the rectus femoris (Pfusterschmied et al., 2013), which may result in less significant activation effects. Furthermore, due to its cross-joint function and superficial anatomical position, the rectus femoris may respond less significantly to unstable loads compared to deeper muscles (Watanabe, Kouzaki & Moritani, 2014). Subgroup analysis results show that the Bosu ball has a negative effect on rectus femoris activation, possibly because using a Bosu ball during squats limits the trainee’s ability to use heavier weights for load training, affecting strength gain (Beck, Stock & DeFreitas, 2012; Drinkwater, Pritchett & Behm, 2007). During exercise, activation of the rectus femoris may be more allocated to synergistic muscles to maintain overall stability, thereby reducing its training load (Landry, Nigg & Tecante, 2010). Under unstable conditions, while the rectus femoris is activated, there is a certain degree of reduction in lower limb force output, weakening overall dynamic stability of the lower limb, and the effects achieved are less significant compared to training under stable load conditions. The article is the first to discover a negative activation effect of using a Bosu ball during squats for individuals with extensive exercise experience. Future research requires more original trials to validate the article’s preliminary findings.

Physiological mechanisms of unstable training in promoting muscle activation

Unstable training is a widely used method to enhance physical fitness and improve neuromuscular control. Its core concept lies in promoting the body’s neuromuscular system to adapt accordingly by exercising in unstable environments, thereby increasing muscle activation levels (Tanimoto et al., 2008). In this process, multiple studies have shown that load is the main variable stimulating superficial muscle activity (Dong et al., 2017; Zhao, Li & Huo, 2021; Qiao et al., 2021), requiring the nervous system to increase the recruitment of motor units (Silva-Batista et al., 2017; Marshall & Murphy, 2005), especially higher threshold motor units (Anderson & Behm, 2004), to maintain body stability. At the same time, when performing training that changes muscle work conditions, it is not only necessary to consider changes in muscle strength activity but also pay attention to changes in exertion patterns (Hong, Liu & Chen, 2016). Among them, the role of the kinetic chain effect, motor muscles, and stabilizing muscles in unstable training should be clearly understood. The kinetic chain is the process by which the body transmits force from the core region to distal joints and muscles through the myofascial connection system to complete coordinated movements (Kaur, Bhanot & Ferreira, 2020). This effect is particularly important in unstable environments, helping to enhance stability and the precision of movements. Stabilizing muscles, through contraction, promote joint stiffness and respond to disturbances early through feedforward or feedback control mechanisms (Sangwan, Green & Taylor, 2014). Unstable training primarily focuses on neural adaptability of muscles, and UT seems to provide better adaptive stimuli for stabilizing muscles than for primary motor muscles (Kibele, 2017). Additionally, unstable movements seem more suitable for weight loss programs, as total energy expenditure is significantly greater when exercising on unstable platforms.

In terms of muscle fiber types, under normal circumstances, the body’s low-threshold type I muscle fibers are preferentially recruited over high-threshold type II muscle fibers, especially in light and low-intensity exercise tasks (Henneman, Somjen & Carpenter, 1965). However, under unstable conditions, the recruitment order of muscle fibers not only follows the size principle (such as preferentially recruiting type I slow fibers) but may also recruit some type II fast fibers due to increased instability, to quickly respond to balance challenges and posture adjustments (Behm et al., 2010). Therefore, in unstable raining, the upper limb muscles exhibit higher levels of type II muscle fiber recruitment in unstable environments (Kowalski et al., 2022), enabling trainees to better cope with external changes.

In terms of proprioception. When the external environment changes, the proprioceptors in the joints, tendons, and muscles (such as muscle spindles and Golgi tendon organs) become more sensitive to help the body perceive position and adjust posture (Proske & Gandevia, 2012). The feedback from the receptors is processed by the central nervous system, further enhancing muscle activation efficiency and neuromuscular coordination. This helps promote motor learning and neuroplasticity (Taube, Gruber & Gollhofer, 2008), directly reflected in the improvement of one’s ability to control muscles.

Limitations of the study

Although this meta-analysis included literature that underwent strict selection and exclusion, there are still certain limitations. First, among the 28 included articles, it was not possible to fully analyze all existing exercise equipment, exercise intensity, and exercise methods, and the coverage of exercise experience and contraction patterns was not complete, with some lacking corresponding cross-analysis. Secondly, there are limitations of heterogeneity and publication bias. The sample of partially included articles is small, and in the subgroup analysis of lower limb muscles, comprehensive and detailed analysis cannot be carried out. Third, in the included literature, there was a lack of separate unified female participants, leading to an inability to analyze differences between men and women in unstable training. Finally, due to the heterogeneity of the trials and individual differences among participants, this could lead to some data having heterogeneity, potentially causing bias in the results.

Practical applications

The article systematically reviewed the effects of unstable training on the activation of different upper limb muscles. Through subgroup and sensitivity analyses, while ensuring the quality of the literature, it explored the effects of unstable training on exercise equipment, exercise intensity, exercise methods, training experience, and contraction patterns, as well as potential regulatory effects, thereby enhancing the credibility and scientific nature of the research results.

Statistically, unstable training significantly improved the activation levels of core muscles (rectus abdominis, internal oblique, external oblique, erector spinae), upper limb muscles (biceps brachii, trapezius, serratus anterior, and triceps brachii), and lower limb muscles (soleus and gluteus medius). For trained individuals, unstable training increases the difficulty of movements to raise the demand for neuromuscular control. Athletes, with rich training experience, can better activate the prime movers in their specialized exercises through unstable training. Moreover, it can strengthen the neuromuscular control in the specialized movement patterns, thus further enhancing their specialized abilities. Individuals with less training experience tend to activate the superficial muscle groups due to insufficient neural control ability. They can strengthen their more basic balance ability and muscle strength through unstable training. Notably, when training on unstable surfaces, the body needs to coordinate a large number of muscle groups and continuously adjust stability to maintain balance, which may lead to unstable blood pressure increases (Amano et al., 2001). For individuals at higher cardiovascular risk (such as those with hypertension or coronary heart disease), the increased demand for neuromuscular control and response during training on unstable surfaces may significantly elevate heart rate and respiratory frequency, placing greater strain on the cardiovascular system and increasing health risks. Additionally, the increased activity of the sympathetic nervous system under unstable training conditions may further lead to elevated blood pressure and accelerated heart rate, increasing the risk of cardiovascular emergencies. For individuals with weak strength, it is advisable to use body weight and take protective measures when choosing unstable training to avoid sports injuries. Finally, for individuals with sports injuries, unstable training mobilizes multiple joints and coordinates the distribution of different motor units, which helps increase muscle activation under lower loads, reduce compensatory movements, and effectively promote functional recovery, enhance core stability, improve neuromuscular control, enhance balance and coordination, and prevent secondary injuries. Therefore, future research should focus on the needs of different populations to determine more accurate selection schemes, integrate them with specialization, and explore the multidimensional value of unstable training.

Conclusion

Unstable training can enhance the activation levels of core muscles (rectus abdominis, internal oblique, external oblique, erector spinae), upper limb muscles (biceps brachii, trapezius, serratus anterior, triceps brachii), and lower limb muscles (soleus, gluteus medius). Although existing studies have verified the advantages of unstable training (such as optimizing the transfer of the kinetic chain, reducing compensatory activation, and improving neuromuscular control, etc.), there are still key limitations and unresolved issues in specific practical applications, especially in terms of exercise intensity. In terms of exercise movements, more functional movements can be combined or created to improve the exercise effect. Future research requires more comprehensive randomized controlled trials to further demonstrate and strengthen these conclusions.

Supplemental Information

Supplemental Information 1 PRISMA checklist.

Supplemental Information 2 Raw Data.

Additional Information and Declarations

Competing Interests

The authors declare that they have no competing interests.

Author Contributions

Zihan Bao conceived and designed the experiments, performed the experiments, analyzed the data, prepared figures and/or tables, authored or reviewed drafts of the article, and approved the final draft.

Shun Wang performed the experiments, authored or reviewed drafts of the article, and approved the final draft.

Ziyang Li analyzed the data, authored or reviewed drafts of the article, and approved the final draft.

Data Availability

The following information was supplied regarding data availability:

This is a systematic review/meta-analysis.

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
