# Peer review of "Effects of unstable training on muscle activation: a systematic review and meta-analysis of electromyographic studies"

_PeerJ, doi:10.7717/peerj.19751_

## Round 0.1 · original submission · Major Revisions

Thank you for submitting this manuscript. As you can see there are lots of positive comments from the reviewers. Overall I think this manuscript is well presented and has some merit for publication. Please see below a number of additional comments from me for consideration.

1. The title clarity should be improved

2. L56 "training on muscle activation in various parts" - this is unclear

3. Search terms should have included
"muscle activ*" to allow for "muscle activity" not just "activation"
electromyography or "EMG"
Please provide some reassurance that important papers were not excluded due to these terms

4. There are some places where the link to the reference is lost "Error! Reference source not found" - please check this prior to submission.

5. There are a lot of figures and tables with this manuscript. My preference would be to move the tables with the individual muscle analysis to supplementary material and along with all of the individual forest plots. You could create a synthesised forest plot of all of the overall effects for all muscles which would be much more informative.


**Language Note:** The review process has identified that the English language must be improved. PeerJ can provide language editing services - please contact us at [email protected] for pricing (be sure to provide your manuscript number and title). Alternatively, you should make your own arrangements to improve the language quality and provide details in your response letter. – PeerJ Staff

·

Basic reporting

no comment

Experimental design

no comment

Validity of the findings

no comment

Additional comments

Thank you for the opportunity to review your manuscript entitled "Effects of Muscle Electromyography Analyses of Unstable Training on Muscle Activation: A Systematic Review and Meta-Analysis". This is an interesting and relevant study that systematically evaluates the effects of unstable training on muscle activation. The manuscript is well-structured and addresses an important topic in sports science. The authors should be commended for their effort in synthesizing the current evidence and providing practical applications for unstable training

With some minor revisions to address the areas for improvement outlined below, I believe the manuscript will be suitable for publication.

Reviewer 2 ·

Basic reporting

This systematic review and meta-analysis examined the effects of unstable training (UT) on muscle activation across different muscle groups. The study is of interest, but several points need to be improved and typing errors and incorrect citation manners need to be corrected. Please find the suggestions below.

Title and abstract
1. Effects of muscle electromyography analyses" is unclear and redundant. EMG is a method for analyzing muscle activation, so it should not be framed as the effect being studied. The title like "Effects of Unstable Training on Muscle Activation: A Systematic Review and Meta-Analysis of Electromyographic Studies" may be better for highlighting what the study's specific focus.
2. "Subgroup analyses were performed on five variables: Exercise Equipment, exercise intensity, exercise mode, exercise experience, and contraction mode." – This sentence needs clearer formatting, especially with inconsistent capitalization.
Introduction
1. Please clarify the definition of unstable training. The definition of unstable training (UT) in the text is somewhat general and could be more precise. While it mentions exercises performed on unstable surfaces to enhance core and surrounding muscle activation, it does not clearly differentiate UT from related concepts such as perturbation training or instability resistance training.
2. In line 39, "Muscle Activation" should not be capitalized mid-sentence.
3. The study does not clearly specify which strength and stability exercises were included under unstable training (UT) in a focused manner. However, from the abstract, it is possible to infer that the review analyzed exercises using various unstable surfaces and equipment. This point should be clarified in the introduction.
4. Please check the citation format shown in the whole manuscript. There are several errors.
5. The introduction does not clearly define or describe the characteristics of TRX suspension training.

Experimental design

Methods
1. The listed databases (CNKI, VIP, Wanfang, PubMed, EBSCOhost, Web of Science) are extensive and inclusive, but it would be valuable to mention any language or geographical limitations, if applicable. Is there a focus on studies published in English, Chinese, or another language?
2. The statement that the search covers the time "from the establishment of each database to November 26, 2024" is somewhat ambiguous. It would be better to specify whether this time frame pertains to the most recent updates or if it is a cut-off date.
3. Please specify the EMG outcomes selected in this study.
4. Please add more detail on how disagreements are resolved (e.g., through consensus, discussion, or arbitration) would provide transparency.
5. The statement "Ten studies reported on training experience" is vague. It would be helpful to specify what is meant by "training experience" (e.g., whether the studies considered participants' prior experience in strength training or specific unstable training exercises). Further elaborating on this could clarify how the training experience was defined and its relevance to the study outcomes.
6. The use of the Cochrane Risk of Bias Assessment Tool is appropriate, but the criteria for classification as "low risk," "high risk," or "unclear risk" could be briefly outlined. This would help readers understand the assessment process.
7. Please check the abbreviation of Randomized crossover trials shown in line 75. It seems incorrect.
8. The study's focus on the effects of unstable training on the activation. It would be helpful to provide more specific details on the EMG variables used to measure this activation.
9. Please check the citation format shown in the whole manuscript. There are several errors.

Validity of the findings

Results
1. The reporting of heterogeneity (e.g., I²=26%, P=0.14) is clear, but a brief explanation of what these values represent could enhance understanding.
2. Please add a brief interpretation of the SMD (Standardized Mean Difference). This would be useful for the reader.
3. Please check consistency in reporting the effects size in bracket (e.g., (7 effects, 101 subjects) in line 176,183, 191, 201, 210, 221, 23, 226, 232, 239, 244, 246, 248 and so on.
4. Please check the copy and paste sentence that the duplicates are shown in lines 142, 150, 158, 166.
5. The text begins by stating that 28 articles were included in the study (line 109), but later it mentions 52 studies (line 116) that were compared for muscle activation between unstable and stable surfaces. Please clarify.
6. Please check the citation format shown in the whole manuscript. There are several errors.
7. In the figure caption of the forest plot, it is important to include additional details about the data presented. Please specify the type of data shown in the plot, such as the effect size, confidence intervals, or other relevant measures. Additionally, clarify which exercises or conditions are being compared in the plot. It would be helpful to explain how the studies are grouped (e.g., by exercise type, equipment used, or any other criteria). Furthermore, please mention the labels for both the x and y axes, as well as what they represent.
8. Figure 20 appears too small, which makes it difficult to view the details clearly. Please consider increasing the size of the figure so that all information is legible and easy to interpret. This will help enhance the clarity of the data presented.
Discussion and conclusion
1. Phrases like "this is consistent with the results of the systematic review by Oliva-Lozano JM et al. [43]" are repeated several times throughout the discussion (e.g., lines 284, 299), which could be avoided by summarizing key comparisons in one section rather than repeating the same point.
2. The discussion jumps from one point to another quite abruptly (e.g., from results on RA activation to IO activation). Adding transition phrases could help guide the reader through the different sections more smoothly.
3. There is limited discussion on how training duration and intensity were controlled across the studies. How might variations in these factors influence the outcomes?
4. According to the sentence “In summary, unstable training can effectively activate the abdominal core muscles,”. It would be good if the authors mentioned which exercise schemes or equipment are most effective for different core muscles could help reinforce the key takeaways from the analysis.
5. The paper mentions that "Swiss balls may lead to negative effects" but doesn’t provide enough evidence or reasoning to explain why this occurs, especially given the previous statement that using Swiss balls might benefit core muscle activation. Please clarify these contradictory statements.
6. According to this sentence "experienced individuals tend to engage more synergistic muscles". Please explain how this relates to other studies or why this phenomenon occurs, which could enhance the depth of the analysis.
7. The conclusion ("trainers should develop a feasible training plan based on their own training experience...") is somewhat vague. It would be helpful to provide more specific recommendations or actionable insights based on the findings.
8. Please check the citation format shown in the whole manuscript. There are several errors.

Reviewer 3 ·

Basic reporting

Language:
The manuscript generally demonstrates professional English. However, certain sections (e.g., Introduction, Results, and Discussion) require language refinement to improve readability and clarity. Specifically, phrasing in the introduction (lines 31-58) should be more precise and concise.

Title & Abstract:
The title is clear but could be more specific regarding population or muscle groups. The abstract succinctly summarizes methods and findings but lacks brief comments on limitations or future research directions.

Introduction & Background:
The introduction clearly states the rationale and objective, situating the review within existing literature. Nonetheless, a more explicit discussion of the existing knowledge gap and clearer justification for the need for this systematic review and meta-analysis are required.

Literature Referencing:
Literature is appropriately referenced and relevant. However, some key recent articles (last two years) might be missing and should be included to enhance context.

Figures & Tables:
Figures and tables provided (especially the PRISMA flow diagram and subgroup analyses tables) are relevant but require formatting improvements for clarity. Specifically, some subgroup tables (Table 2, Table 3) could benefit from clearer column labeling and alignment.

Raw Data:
Provided clearly, satisfying the PeerJ policy.

Experimental design

Research Question:
The research question is clear, relevant, and meaningful. It directly addresses how unstable training impacts muscle activation compared to stable training.

Methods & Rigor:
Methods are described clearly, including databases searched and software used. However, the manuscript should elaborate on the process of independent screening and data extraction by reviewers to clarify reproducibility further.

Eligibility Criteria:
Criteria for inclusion/exclusion are adequately explained, but further details regarding the exclusion criteria (particularly reasons for exclusion) could improve clarity.

Bias Assessment:
The method of bias assessment using the Cochrane Risk of Bias Tool is appropriate and clearly presented. However, more detailed explanation about how discrepancies were resolved is needed.

Validity of the findings

Robustness of Results:
The meta-analysis is rigorous, and the subgroup analyses significantly enhance understanding. However, the high heterogeneity found in some muscle groups (e.g., erector spinae) needs clearer explanation and discussion of potential reasons or biases influencing these results.

Statistical Analyses:
The use of statistical tools (RevManager, Stata, R) is well-justified and appropriate. Sensitivity and subgroup analyses are adequately used, but additional meta-regression analysis could enhance the understanding of factors influencing heterogeneity.

Discussion and Interpretation:
The discussion needs significant expansion. Interpretations of subgroup findings should be expanded to include more explicit clinical or practical implications. Furthermore, inconsistencies or unexpected findings (negative activation effects) should be explored more thoroughly.

Limitations:
Limitations of the included studies and the systematic review itself are underreported. Clearly articulated limitations concerning study heterogeneity, small sample sizes in some subgroups, and publication bias should be thoroughly discussed.

Conclusions:
Conclusions clearly reflect the findings but must acknowledge the limitations more explicitly. More specific recommendations for future research directions and practical application in rehabilitation or sports contexts are necessary.

Additional comments

The manuscript addresses an important area in sports science and rehabilitation with potential to inform training guidelines significantly. The strength of the manuscript is the rigorous methodological approach, including detailed subgroup analyses.

However, major revisions are required, particularly enhancing the introduction's context, refining methodological transparency, addressing high heterogeneity explicitly, expanding discussion of implications, and clearly outlining limitations and directions for future research.

Recommended Revisions (Major):

Clarify and explicitly state the research gap in the introduction.

Improve readability and clarity throughout, especially in abstract, introduction, and discussion sections.

Detail the reviewer process of article selection and data extraction, including resolution methods for disagreements.

Clearly explain and discuss high heterogeneity observed in specific muscle group analyses.

Expand the discussion section, explicitly detailing the clinical implications and the relevance of findings for practitioners and researchers.

Include a robust discussion of the limitations of your study and provide explicit recommendations for future research directions.

Conclusion:
The manuscript is promising, but substantial revisions are required before acceptance. Attention to these detailed suggestions will significantly strengthen the manuscript, improving its clarity, readability, and scientific rigor.

---

## Round 0.2 · Minor Revisions

I agree with the reviewers that the manuscript is much improved. Please address the final comments of reviewer 2 with respect to (1) the consistency of citation formatting, (2) the clarity of figure presentation, and (3) the need for more detailed figure captions.

·

Basic reporting

Dear Editors,

Thank you for the opportunity to review the revised manuscript entitled "Effects of unstable training on muscle activation: A Systematic Review and Meta-Analysis of Electromyographic Studies"

I appreciate the authors' thorough and constructive response to the previous round of peer review. The revised version of the manuscript reflects substantial improvements.

I am pleased to recommend this manuscript for publication in its current form

Experimental design

no comment

Validity of the findings

no comment

Reviewer 2 ·

Basic reporting

I have reviewed the revised manuscript and confirm that the authors have effectively addressed the majority of my comments. The definitions of unstable training, methodological details, subgroup analyses, and practical implications have all been significantly improved and are now clearly presented and well-supported.
While a few minor issues remain—particularly in the consistency of citation formatting, the clarity of figure presentation, and the need for more detailed figure captions—these do not compromise the overall scientific quality and integrity of the work. I commend the authors for their substantial and thoughtful revisions.
I support the publication of this manuscript, pending final polishing of these minor issues.

Experimental design

The experimental design of this systematic review and meta-analysis is generally appropriate and well-executed, particularly for a study of this nature.

Validity of the findings

The validity of the findings in this systematic review and meta-analysis is strong overall, with evidence-based conclusions that are consistent with the reported data.

Additional comments

-

---

## Round 0.3 · accepted · Accept

Thank you for addressing the review comments I am happy to recommend publication. Congratulations.

Reviewer 2 ·

Basic reporting

The manuscript is well-written and clearly organized, with substantial improvements since the previous version.

Experimental design

The study adheres to PRISMA guidelines and is registered with PROSPERO. It presents a clear research question: how does unstable training influence muscle activation across body regions, as assessed by electromyography (EMG)? The methodology is sound. The inclusion/exclusion criteria are appropriate and follow the PICOS framework. EMG outcomes are well defined (RMS amplitude normalized by MVIC/MVC), and subgroup analyses are logically structured by equipment, intensity, mode, training experience, and contraction type.

Validity of the findings

The results are well supported by statistical evidence. Subgroup effects are clearly reported, and the discussion explores possible mechanisms, including neuromuscular adaptation and proprioceptive control. The authors acknowledge the limitations of their meta-analysis, including variation in study designs and sample sizes.